# Neutral vs. non-neutral genetic footprints of *Plasmodium falciparum* multiclonal infections

**Frédéric Labbé**[1], **Qixin He**[2], **Qi Zhan**[1], **Kathryn E. Tiedje**[3,4], **Dionne C. Argyropoulos**[3,4], **Mun Hua Tan**[3,4], **Anita Ghansah**[5], **Karen P. Day**[3,4], **Mercedes Pascual**[1,6]*

**1** Department of Ecology and Evolution, The University of Chicago, Chicago, Illinois, United States of America, **2** Department of Biological Sciences, Purdue University, West Lafayette, Indianapolis, United States of America, **3** School of BioSciences, Bio21 Institute, The University of Melbourne, Melbourne, Australia, **4** Department of Microbiology and Immunology, Bio21 Institute and Peter Doherty Institute, The University of Melbourne, Melbourne, Australia, **5** Department of Parasitology, Noguchi Memorial Institute for Medical Research, College of Health Science, University of Ghana, Legon, Ghana, **6** Santa Fe Institute, Santa Fe, New Mexico, United States of America

* pascualmm@uchicago.edu

**Data Availability Statement:** The agent-based stochastic simulator of malaria dynamics and the processing scripts to reproduce all the figures are stored and annotated on GitHub: https://github.

## Abstract

At a time when effective tools for monitoring malaria control and eradication efforts are crucial, the increasing availability of molecular data motivates their application to epidemiology. The multiplicity of infection (MOI), defined as the number of genetically distinct parasite strains co-infecting a host, is one key epidemiological parameter for evaluating malaria interventions. Estimating MOI remains a challenge for high-transmission settings where individuals typically carry multiple co-occurring infections. Several quantitative approaches have been developed to estimate MOI, including two cost-effective ones relying on molecular data: i) THE REAL McCOIL method is based on putatively neutral single nucleotide polymorphism loci, and ii) the *var*coding method is a fingerprinting approach that relies on the diversity and limited repertoire overlap of the *var* multigene family encoding the major *Plasmodium falciparum* blood-stage antigen PfEMP1 and is therefore under selection. In this study, we assess the robustness of the MOI estimates generated with these two approaches by simulating *P. falciparum* malaria dynamics under three transmission conditions using an extension of a previously developed stochastic agent-based model. We demonstrate that these approaches are complementary and best considered across distinct transmission intensities. While *var*coding can underestimate MOI, it allows robust estimation, especially under high transmission where repertoire overlap is extremely limited from frequency-dependent selection. In contrast, THE REAL McCOIL often considerably overestimates MOI, but still provides reasonable estimates for low and moderate transmission. Regardless of transmission intensity, results for THE REAL McCOIL indicate that an inaccurate tail at high MOI values is generated, and that at high transmission, an apparently reasonable estimated MOI distribution can arise from some degree of compensation between overestimation and underestimation. As many countries pursue malaria elimination targets, defining the most suitable approach to estimate MOI based on sample size and local transmission intensity is highly recommended for monitoring the impact of intervention programs.

com/pascualgroup/varmodel3. The SNP data used for this analysis are available in Dryad at https://doi.org/10.5061/dryad.jsxksn0bp. The DBLα sequences used for this analysis are available in GenBank under BioProject Number: PRJNA396962.

**Funding:** This research was initially supported by the Fogarty International Center at the National Institutes of Health (Program on the Ecology and Evolution of Infectious Diseases), grant R01-TW009670 to KD and MP. Funding was provided by the joint NIH-NSF-NIFA Ecology and Evolution of Infectious Disease award R01-AI149779 to KD and MP. The funders had no role in study design, data collection and analysis, decision to publish, or preparation of the manuscript.

**Competing interests:** The authors have declared that no competing interests exist.

## Author summary

Despite control and elimination efforts, malaria continues to be a serious public health threat especially in high-transmission regions. Molecular tools for evaluating these efforts include those seeking to estimate multiplicity (or complexity) of infection (MOI), the number of genetically distinct parasite strains co-infecting a host, a key epidemiological parameter. MOI estimation remains challenging in high-transmission regions where hosts typically carry multiple co-infections by *Plasmodium falciparum*. THE REAL McCOIL and the *var*coding are two cost-effective methods relying on distinct parts of the parasite genome, those respectively under neutrality and selection. The more recent *var*coding approach relies on the *var* multigene family encoding for the major blood-stage antigen and contributing to a complex immune evasion strategy of the parasite. We compare the performance of the two methods by simulating disease dynamics under different transmission intensities with a stochastic agent-based model tracking infection by different parasite genomes and immune memory in individual hosts, then sampling resulting infections to estimate MOI. Although THE REAL McCOIL provides reasonable estimates for low and moderate transmission, *var*coding allows more robust estimates especially under high transmission. Regardless of transmission intensity, THE REAL McCOIL generates an estimated MOI distribution with a tail of high values that is absent in the simulations, and at high transmission, it produces a distribution that can appear reasonable to some degree for the wrong reason, from over and under-estimated values compensating each other. Defining the most suitable approach to estimate MOI based on local transmission intensity is highly recommended for hyper-diverse pathogens such as *P. falciparum*.

## Introduction

Malaria deaths have steadily and significantly declined over the period 2000–2019 in response to control and elimination efforts [1]. However, malaria continues to be a serious threat causing approximately half a million deaths in 2019, especially among young children in high-transmission endemic regions in Africa. In these regions, infections are characterized by multiple genetically distinct *Plasmodium* parasite genotypes simultaneously infecting a host. Multiplicity of infection (MOI), also known as complexity of infection (COI), is defined as the number of genetically distinct parasite strains co-infecting a single host [2]. Multiclonal infections (i.e., MOI > 1) can be the result of a single bite by a mosquito transmitting more than one genetic parasite strain or independent bites by infected mosquitoes (also termed superinfection). The number of co-infections is associated with transmission intensity, clinical risk, age and immunity [3–6].

Given the potential relevance of MOI to malaria surveillance, various approaches have been developed to estimate MOI from clinical samples. As *Plasmodium falciparum* parasites reproduce asexually as haploid stages when they infect humans, signatures of polymorphic genotypes are evidence of multiclonal infections. While any highly polymorphic marker is thus suitable for estimating MOI, it remains a challenge to accurately measure MOI in malaria-endemic areas where multiclonal infections are common. The most common approach for determining MOI involves size-polymorphic antigenic markers, such as *msp1*, *msp2*, *msp3*, *glurp*, *ama1*, and *csp*, that can be amplified by PCR and determined by capillary electrophoresis or agarose gel [7]. Similarly, microsatellites, also termed simple sequence repeat (SSR), are another type of size-polymorphic marker that can be amplified by PCR to estimate MOI by

determining the number of alleles detected [5,8–12]. However, despite improved resolution of allele detection by capillary electrophoresis, these approaches based on size-polymorphisms usually involve a certain degree of subjective interpretation, are unable to discriminate alleles of similar sizes [13], and create PCR artifacts resulting in inconsistent results, especially for MOI > 5 [14,15]. As an alternative to size-polymorphic markers, other methods of determining MOI have focused on reconstructing haplotypes from genotyping or sequencing data (e.g., estMOI, FWS, and DEploid) [16–19]. Whole-genome sequencing is currently not a cost-effective approach when MOI is the main interest of a study, and these haplotype-reconstruction approaches are computationally intensive, resulting in limited MOI reliability for highly complex infections [20]. Finally, two cost-effective molecular approaches, known as THE REAL McCOIL [21] and *var*coding [22,23], have been more recently developed to identify and track MOI with standard laboratory equipment. They differ in important ways as they rely on contrasting parts of the genome, respectively under neutrality and immune selection.

As many possible genotypes exist among a combination of several genome-wide single nucleotide polymorphisms (SNPs), methods of determining MOI have focused on neutral SNP data to discriminate among strains [24]. Galinsky *et al.* [25] developed the COIL approach to estimate MOI from a panel of bi-allelic SNP data, but this method relies on monoclonal infections (MOI = 1) for estimation of allele frequencies or requires external allele frequency data. As external allele frequency data may only be available for specific locations and be heterogeneous in space and time, an analytical approach, called THE REAL McCOIL (Turning HEterozygous SNP data into Robust Estimates of ALelle frequency, via Markov chain Monte Carlo, and Complexity Of Infection using Likelihood), was developed to simultaneously estimate the MOI within a human host and the allele frequencies in the population based on a panel of SNPs [21]. Using simulations and 105 SNP data from cross-sectional surveys in Uganda, Chang *et al.* (2017) showed that THE REAL McCOIL approach improved performance in estimates of both quantities, despite the uncertainty of these estimates increasing with true MOI.

The more recent *var*coding approach (also termed *var* genotyping or *var* fingerprinting) [22,23], employs the highly polymorphic sequences encoding the immunogenic DBLα domain of PfEMP1 (*Plasmodium falciparum* erythrocyte membrane protein 1), the major surface antigen of the blood stage of infection [26]. During an infection, PfEMP1 molecules are exported by the parasite to the surface of the infected erythrocytes, where they influence virulence of the disease and become a target of the adaptive immune system [27]. The multigene family known as *var* encodes variants of this surface antigen which can reach tens of thousands of variants in endemic populations [22,28–33]. The sequential expression of a set of up to 60 *var* genes per parasite (hereafter, a repertoire) leads to immune evasion, prolongs infection duration, and establishes chronic infections enabling onward transmission [34,35]. Immune evasion is particularly important in high-transmission regions where *var* repertoires are composed of largely distinct sets of *var* genes [22,28–31,36]. This non-random composition of *var* repertoires has been shown to result from negative frequency-dependent immune selection [30,36]. Largely non-overlapping *var* repertoires enhance survival in semi-immune hosts, in accordance with earlier models of parasite competition for hosts via specific immunity [37] and more recent deep molecular sampling of local populations and computational theory [22,30,36]. This feature of *var* population structure allows distinct strains to accumulate in the blood of human hosts. The extensive diversity of the *var* gene family together with the very low percentage of *var* genes shared between parasites facilitate measuring MOI by amplifying, pooling, sequencing, and counting the number of DBLα types in a host. This feature of *var* population genetics is the basis of the fingerprinting concept of *var*coding. From a single PCR with degenerate primers and amplicon sequencing, the method counts unique DBLα types per infection. It is

not based on assigning haplotypes but instead, it assumes a set number of types per genome based on control data accounting for PCR sampling errors to calculate MOI [22,23].

As *var*coding does not require haplotype construction, we propose that this method is particularly well suited for high transmission where THE REAL McCOIL has shown some limitations due to the bi-allelic nature of the SNP calling [21]. To evaluate the relative performance of these two contrasting approaches to estimate MOI across different transmission settings, this study simulates malaria transmission under low, moderate, and high transmission using an extended agent-based model (ABM). We specifically extend a previously developed stochastic computational model to incorporate neutral bi-allelic SNPs which, together with the *var* genes, can be used for MOI estimation. Depending on transmission intensity, we ask whether one of these approaches is more accurate than the other. When most infections are multiclonal (i.e., MOI > 1), we demonstrate that THE REAL McCOIL and the *var*coding approaches tend to overestimate and underestimate the MOI, respectively. Moreover, while the high diversity of the *var* gene family allows robust MOI estimation with the *var*coding approach, especially across high-transmission settings, THE REAL McCOIL provides reasonable estimates across low- and moderate-transmission settings where the *var*coding can be limited by partially overlapping *var* repertoires. We discuss the limitations and advantages of these two approaches to determine the multiplicity of malaria parasite infection as well as their implications for malaria surveillance.

## Results

Accurate estimation of multiplicity of infection is important for evaluating current intervention strategies against malaria and thus defining or adapting future ones. We evaluated two recently developed MOI estimation approaches by simulating malaria transmission using an extended ABM, and sampling, estimating, and comparing MOI under three different transmission settings (i.e., "low", "moderate", or "high").

For each simulation under low-, moderate-, or high-transmission intensity, 2000 individuals were sampled at the end of the wet season (S1 Fig). On the one hand, the number of sampled individuals per age class is consistent with the age distribution and the size of each age class (S1A and S1B Figs). On the other hand, the number of infected sampled individuals was significantly and negatively correlated with the age of the hosts, as expected (Pearson correlation test; high-transmission: r = -0.377, *P*-value < 2.2e-16; moderate-transmission: r = -0.240, *P*-value = < 2.2e-16; low-transmission: r = -0.005, *P*-value = 1.1e-3). The hosts between 0 and 5 years old thus exhibit the lowest number of infections with an average of 3 ± 2 (mean ± SD), 23 ± 5, and 33 ± 6 sampled hosts for the low-, moderate-, and high-transmission simulations, respectively (S1C Fig). As expected, the number of sampled hosts is higher for simulations with high-transmission settings (1239 ± 22) than for those with moderate (677 ± 31) and low-transmission settings (167 ± 24) (S1C Fig). Consistently, the average EIR and prevalence were also higher for simulations with high-transmission settings than for those with low- or moderate-transmission settings (S2 Fig). For instance, low-transmission setting simulations have an average of 0.25 ± 0.04 infectious bites per host per year and 0.08 ± 0.01 infected cases, whereas moderate-transmission simulations have an average of 5.77 ± 0.22 infectious bites per host per year and 0.34 ± 0.02 infected cases, and high-transmission simulations have an average of 21.88 ± 0.21 infectious bites per host per year and 0.62 ± 0.01 infected cases, reflecting highly contrasting malaria transmission intensities. We note that in our simulations "infectious bites" were computed as infectious contact events experienced by a host, and that for generality purposes, we set the transmissibility probability specifying whether such contact results in infection to 0.5 (S1 Table). Thus, for comparison purposes with empirical values, the EIR values

obtained in the simulations should be divided by such probability. Consistently with these epidemiological and genetic diversity statistics, the true MOI distribution generated by the simulations is also significantly different among the three levels of transmission (t-test: *P*-value < 2.2e-16; average true MOI of 1.05 ± 0.21, 1.50 ± 0.82, and 3.81 ± 3.29 for the low-, moderate-, and high-transmission simulations, respectively) (Fig 1B). The true MOI defined in this study corresponds to the number of co-infections, no matter whether the parasites in the different infections are genetically similar or not.

## Estimates with THE REAL McCOIL approach

THE REAL McCOIL approach based on neutral SNP data tends to overestimate MOI when true values range from 3 to 14 (Fig 1A). In contrast, this approach tends to underestimate MOI for true MOI above ~14, values only found in hosts without any single minor allele calls (Figs 2A and 3). Underestimated MOI can differ from the true MOI by up to 1, 3, and 9 co-infections for the low-, moderate-, and high-transmission simulations, respectively. Interestingly, while most hosts with MOI < 3 show accurate MOI estimates, some can differ from the true MOI by up to 18, 12, and 12 co-infections for the low-, moderate-, and high-transmission simulations, respectively. The inaccuracy of the MOI estimates based on THE REAL McCOIL approach, defined as the absolute differences between estimated and true MOI per host, is significantly and positively correlated with the true MOI (*P*-values < 2.2e-16; r = 0.10, r = 0.57, and r = 0.56 for the low-, moderate-, and high-transmission simulations, respectively). Despite the fact that a high proportion of the 95% credible intervals contained the true MOI for the distinct transmission settings (86% ± 35%, 96% ± 21%, and 97% ± 18% for the low-, moderate-, and high-transmission simulations, respectively), the interval sizes covary with the true MOI (r = 0.80, *P*-value < 2.2e-16) and transmission intensity (0.3 ± 1.6, 0.7 ± 2.2, and 3.6 ± 5.2 for the low-, moderate-, and high-transmission simulations, respectively; S4 Fig). The size of this 95% credible intervals can reach up to 19, 18, and 17 co-infections for the low-, moderate-, and high-transmission simulations, respectively.

As the proportion of double allele calls (DACs) per host is significantly and positively correlated with the true MOI (*P*-value < 2.2e-16, r = 0.82) (Figs 3 and S3), inaccuracy is also significantly and positively correlated with the proportion of DACs per host (*P*-value < 2.2e-16, r = 0.59). Consistently, inaccuracy is significantly and negatively correlated with the proportion of single allele calls (minor and major alleles) per host SNP haplotype (*P*-value < 2.2e-16, r = -0.59) (Figs 3 and S3). Inaccuracy is significantly and negatively correlated with the number of SNPs (*P*-value < 2.2e-16, r = -0.15), but estimated MOI with the highest number of SNPs (105 SNPs) can still differ from the true MOI by up to 18 (S5A Fig). Surprisingly, the MOI estimates show higher accuracy for simulations generated with distinct initial SNP allele frequencies but did not seem influenced by the presence of linked SNP loci (S6 Fig). Overall, despite including slightly fewer hosts with low MOI and significantly more hosts with high MOI (Fig 1B), the average estimated MOI was quite similar to that of the true MOI (*P*-values < 2.2e-16; average estimated MOI of 1.36 ± 2.05, 1.70 ± 1.50, and 4.72 ± 4.57 for the low-, moderate-, and high-transmission simulations, respectively). However, due to the combination of overestimated and underestimated MOI values, the distribution showed a second peak of high density around an MOI of 15, which is absent from the true MOI distribution. This peak can correspond to a maximum of ~200 hosts in some simulations.

Consideration of a SNP measurement model, which accounts for potential genotyping failures by randomly replacing some SNP genotypes with missing values, only slightly decreases the accuracy of the MOI estimates based on THE REAL McCOIL approach, relative to MOI estimates made without the measurement model (Figs 1A and 2).

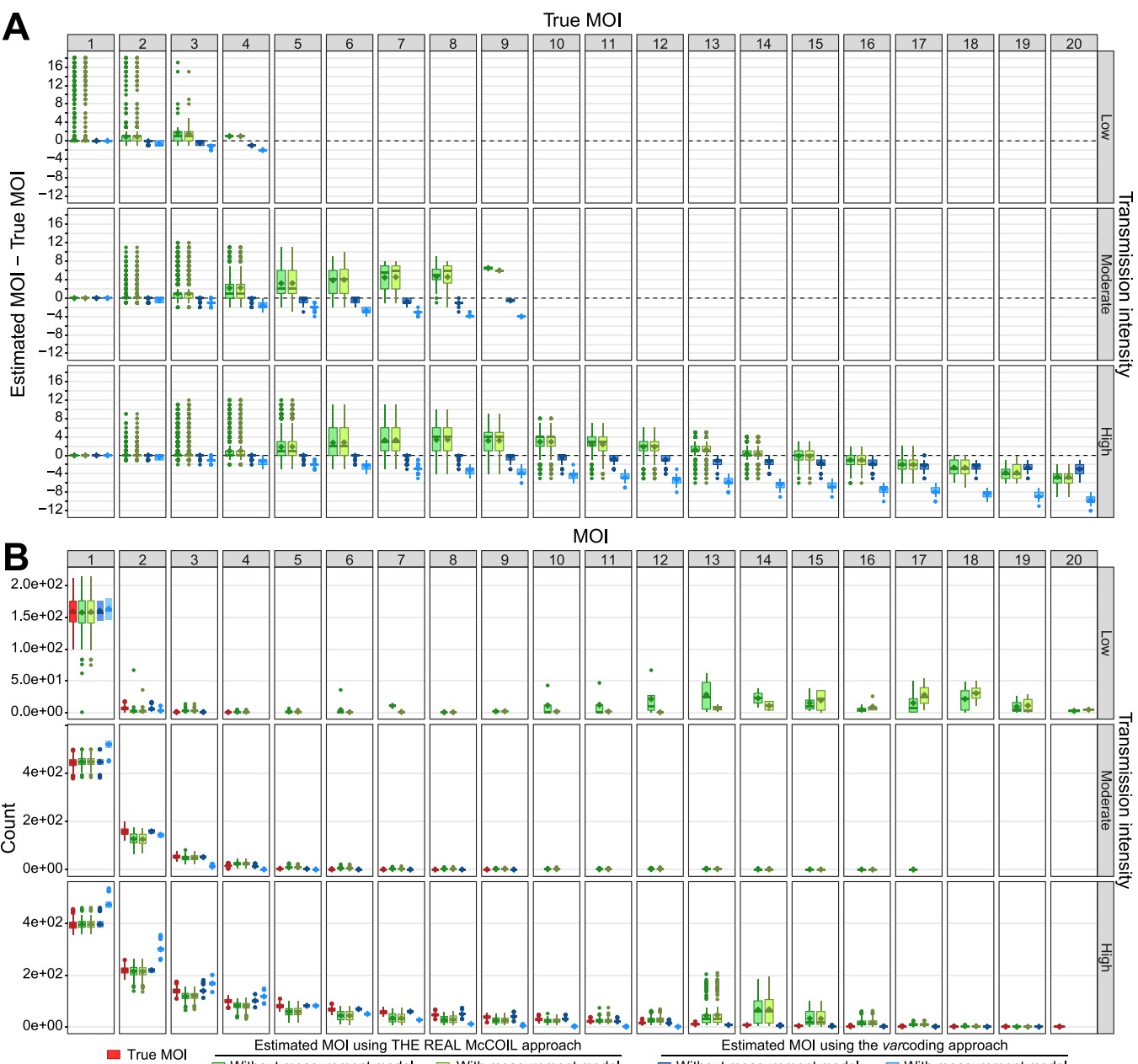

**Fig 1. Multiplicity of infection (MOI).** For each category, the horizontal central solid line represents the median, the diamond represents the mean, the box represents the interquartile range (IQR) from the 25th to 75th centiles, the whiskers indicate the most extreme data point, which is no more than 1.5 times the interquartile range from the box, and the dots show the outliers, i.e., the points beyond the whiskers. The upper, middle, and lower row panels show correspond to simulations under low-, moderate-, and high-transmission settings, respectively (S1 and S2 Tables). **A)** Accuracy of MOI estimates, defined as the difference between estimated and true MOI per host. While null values highlight accurate MOI estimates (indicated by a dashed black horizontal line), the positive and negative values highlight over- and under-estimation, respectively. Estimates with the neutral SNP-based approach (THE REAL McCOIL) are indicated in green, and those with the *var* gene-based approach (*var*coding) are indicated in blue. The dark and light green or blue colors indicate respectively MOI estimations made without and with a measurement model (Fig 2). The column panels show differences for specific true MOI values. **B)** Population distribution of the estimated and true MOI per host from the simulated "true" values and those estimated with the methods indicated by the colors similar to panel A. For high transmission, the distribution obtained with THE REAL McCOIL shows a more pronounced tail than that from the simulated infections, with a secondary peak around MOI = 14. Note that the method considerably over-estimates individual MOI below that value but then under-estimates above it (panel A). Thus, these opposite trends compensate each other to some extent in the population distribution, producing nevertheless a deviation at high values. The *var*coding method provides a good representation of the "true" distribution from the simulations, and of the individual values in general, with a consistent tendency to underestimate when sampling error is taken into account.

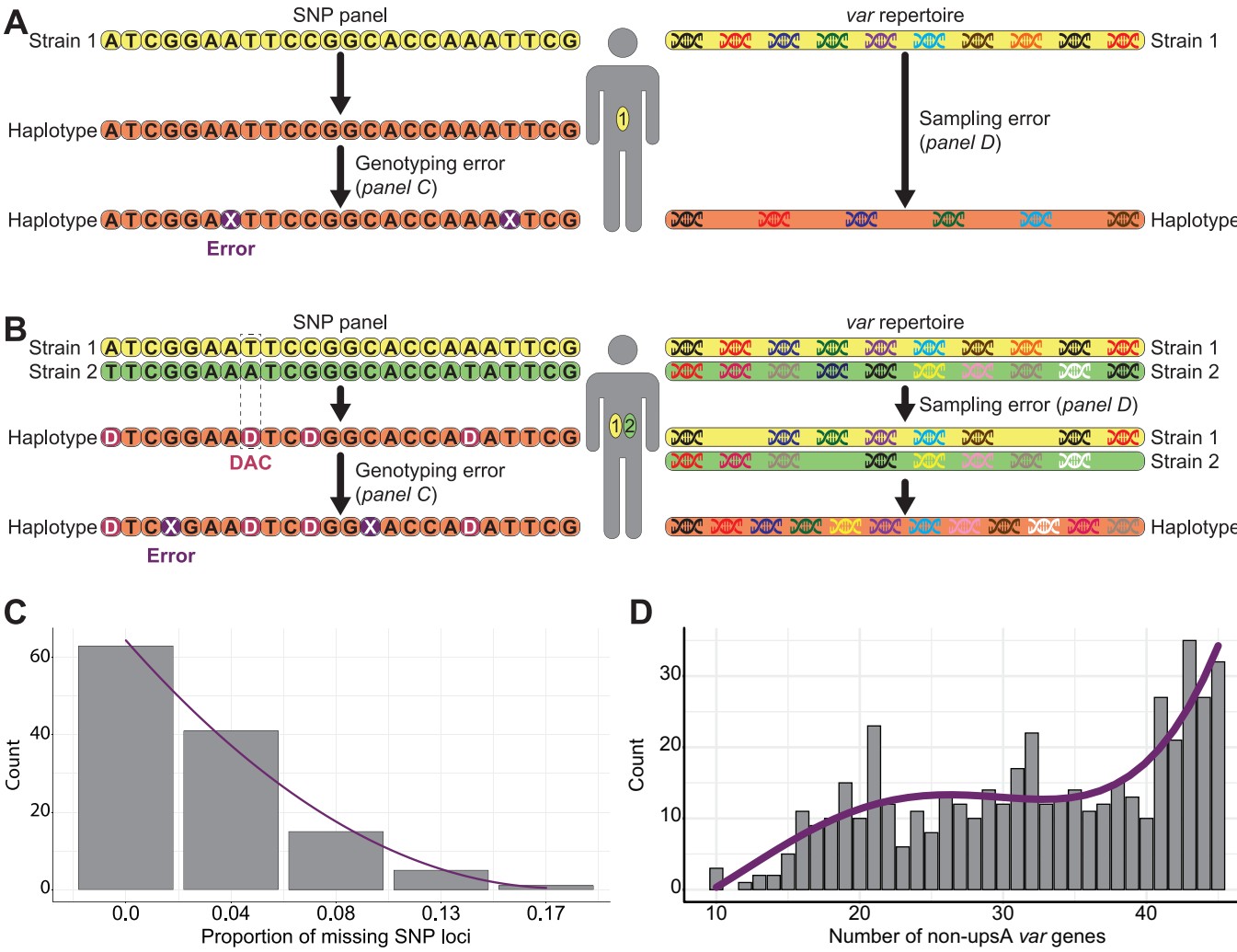

**Fig 2. Measurement models. A)** and **B)** Schematic diagrams of the SNP and *var* measurement models for a host infected by one (MOI = 1) or two (MOI = 2) genetically distinct *P. falciparum* strains, respectively. To account for potential SNP genotyping failures, we randomly replaced the host genotypes with missing data (X). This replacement was implemented by using the distribution illustrated in panel C. When MOI is high, the frequency of double allele calls (DACs) is also high (Figs 3 and S3). To account for *var* gene potential sequencing errors, we sub-sampled the number of *var* genes per repertoire. This sub-sampling was implemented by using the distribution illustrated in panel D. For simplicity, the *var* repertoire in these two examples only consists of 10 *var* genes despite each migrant parasite genome consists of a repertoire of 45 *var* genes in the simulations (S1 Table). **C)** Histogram of the proportion of missing SNP loci per host haplotype from a panel of 24 bi-allelic SNP loci. The genotypes were previously obtained from monoclonal infections sampled during one cross-sectional survey made in 2015 in the Bongo District, in northern Ghana. The purple curves show the best curves that fit the data using the adjusted R-squared. **D)** Histogram of the number of non-upsA (i.e., upsB and upsC) DBLα *var* gene types per repertoire. The molecular sequences were previously sequenced from monoclonal infections, i.e., hosts infected by a single *P. falciparum* strain (MOI = 1), sampled during six cross-sectional surveys made from 2012 to 2016 in the Bongo District, in northern Ghana.

Subsampling the individuals from 2000 to 500 did not reduce the accuracy of the estimated MOI with or without the measurement model (S7 Fig). However, subsampling to 200 individuals significantly increased the number of SNP loci with a MAF < 10%, especially for the low- (8 ± 11 loci) and moderate-transmission setting simulations (4 ± 5 loci). Consequently, due to the high number of SNP loci, and thus individuals, that could not be considered in THE REAL McCOIL analysis, MOI could not have been estimated for most of the low-transmission set-ting simulations when a subsampling of 200 individuals was applied (30%; S8 Fig). Moreover, while THE REAL McCOIL approach could provide MOI estimates for subsamples of the

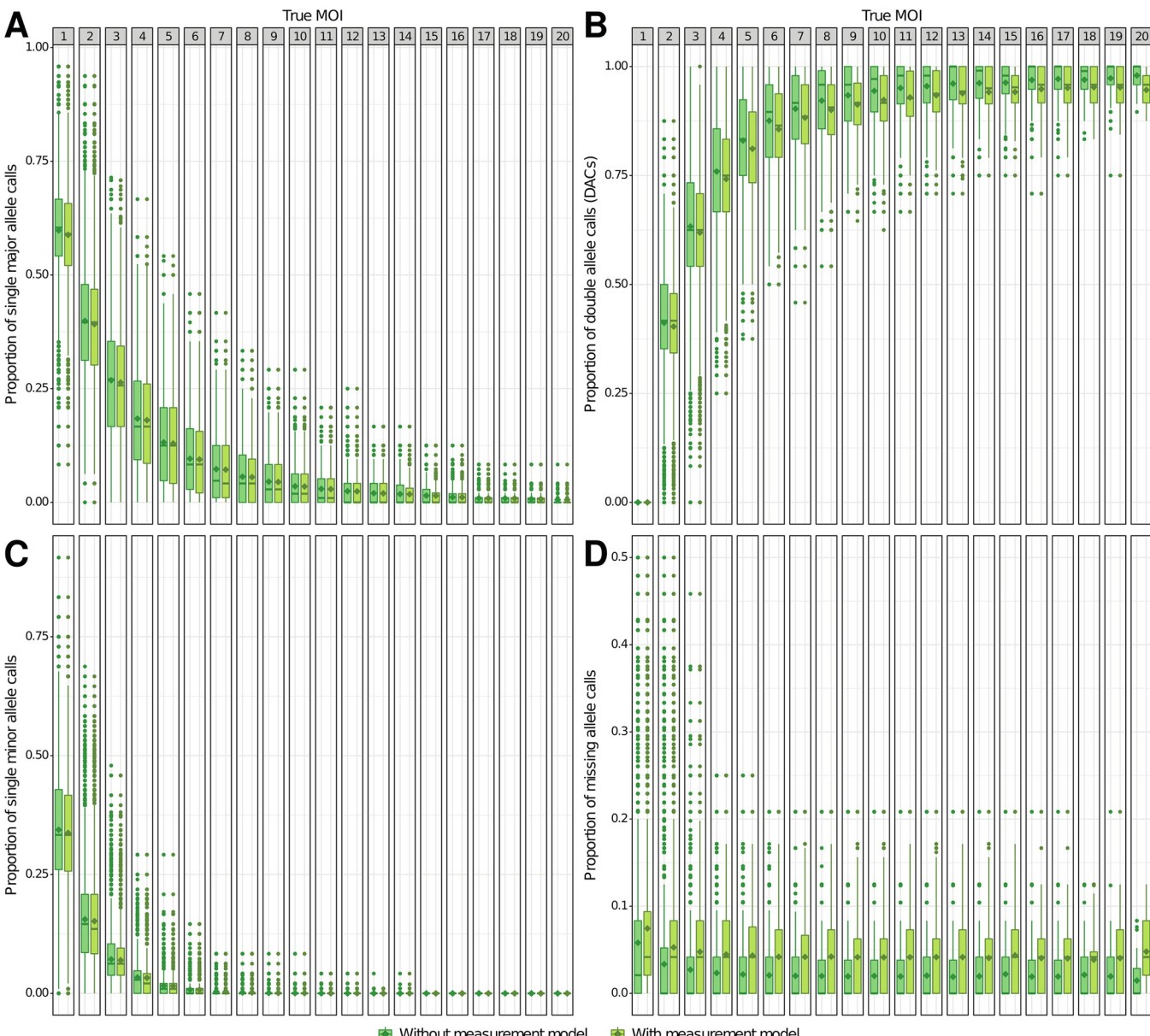

**Fig 3. Proportion of SNP calls (genotypes) per host SNP haplotype. A)** Single major allele calls; **B)** Double allele calls (DACs); **C)** Single minor allele calls; **D)** Missing allele calls. Each SNP call proportion was calculated using the low-, moderate-, and high-transmission setting simulations (S1 and S2 Tables). The column panels show the proportions for specific true MOI values. The dark and light green colors indicate the proportion of calls made without and with a measurement model, respectively (Fig 2). For each category, the horizontal central solid line represents the median, the diamond represents the mean, the box represents the interquartile range (IQR) from the 25th to 75th centiles, the whiskers indicate the most extreme data point, which is no more than 1.5 times the interquartile range from the box, and the dots show the outliers, i.e., the points beyond the whiskers.

moderate- and high-transmission setting simulations, the subsampling of 200 individuals significantly reduced the accuracy of these MOI estimates made with or without the measurement model. Conversely, sampling through several time-points during the malaria season did not influence the performance of THE REAL McCOIL approach (S9 and S10 Figs). Interestingly, simulations involving a background clearance due to processes not explicitly modelled showed a reduction in the accuracy of the estimated MOI, especially for the lowest true MOI values (S9 and S11 Figs). Moreover, the accuracy of MOI estimate for the simulations

involving a temporary intervention remains similar for the moderate- and high-transmission settings. However, as the intervention significantly reduced the prevalence (0.004 ± 0.003) and a SNP required a minimum number of 20 hosts to be considered, THE REAL McCOIL approach was never able to estimate MOI when an intervention was applied to the low-transmission settings (S12 Fig).

Simulations under low- and moderate-transmission settings show more accurate MOI estimates than the simulations under high-transmission settings (Fig 1A). The estimated probability of calling double allele loci single allele loci (e2) was higher than the actual error rate (0.06 ± 0.006 and 0.06 ± 0.007 for the low- and moderate-transmission settings, respectively). For the low-transmission setting, the less accurate MOI estimates corresponded to hosts with a true MOI of 1 (i.e. proportion of DACs = 0%; Figs 1 and 3) which belong to nine simulations (i.e. two replicates of runs 12 and one replicate of runs 6, 10, 15, 18, 20, 21, and 22; Fig 1A; S2 Table). These hosts showed a high proportion of missing calls (e.g. 37.7% ± 12.0% when the MOI estimates differ from the true MOI by 13 co-infections) and/or a higher proportion of single minor allele calls (e.g. 54.7% ± 2.5% when the MOI estimates differ from the true MOI by 18 co-infections) than single major allele calls (e.g. 32.9% ± 7.4% when the MOI estimates differ from the true MOI by 18 co-infections), which may thus contribute to highly inaccuracy estimates despite the absence of DACs. These results could also be explained by highly inaccurate estimated MAF. Indeed, the inaccuracy of the estimated MOI was significantly and positively correlated with the inaccuracy of the estimated MAF per simulation, defined as the sum of the absolute differences between estimated and true MAF per SNP locus ($P$-values < 2.2e-16; r = 0.21 and r = 0.214 for estimates made with or without measurement model, respectively). Interestingly, although THE REAL McCOIL approach generated very inaccurate MAF estimates for a few low-transmission simulations, it typically produced very accurate estimates of the MAF regardless of transmission settings (S13 Fig). Consistent with the MOI results, the accuracy of the MAF estimates per simulation also increases with the number of SNP loci. The inaccuracy of THE REAL McCOIL MAF estimates per locus, defined as the absolute differences between estimated and true MAF per locus, is significantly and negatively correlated with the true MAF, the proportion of DACs, and the proportion of single minor allele calls per locus (S3 Table). Moreover, this inaccuracy of the MAF estimations is also significantly but positively correlated with the proportion of missing allele calls and single major allele calls per locus (S3 Table). When only keeping simulations with the lowest estimated MAF inaccuracy per simulation (inaccuracy < 0.39 = 1$^{st}$ quantile), THE REAL McCOIL still tends to overestimate MOI when true values range from 3 to 14 and to underestimate MOI for true MOI above ~14 (S14 Fig), suggesting that other sources are at the origin of the estimated MOI inaccuracy.

### Estimates with the *var*coding approach

Consistently with an increasing probability of overlapping *var* repertoires for hosts with MOI above 1, the *var*coding approach, which uses the number of *var* genes to estimate MOI, tends to slightly underestimate the MOI (Fig 1A). Its inaccuracy, defined as the absolute difference between estimated and true MOI per host, is significantly and positively correlated with the true MOI ($P$-values < 2.2e-16; r = 0.36, r = 0.29, and r = 0.59 for the low-, moderate-, and high-transmission simulations, respectively). Therefore, MOI values estimated for the hosts with the highest true MOI (i.e., 4, 9, and 20 co-infections for the low-, moderate-, and high-transmission simulations, respectively), differ from the true MOI by up to 1, 1, and 6 co-infections for the low-, moderate-, and high-transmission simulations, respectively. Overall, despite exhibiting slight deviations, with more hosts at low MOI and fewer hosts at high MOI, the distribution of the estimated MOI based on the *var*coding approach is quite similar to that of the

true MOI (*P*-value = 0.8, average estimated MOI of 1.04 ± 0.20; *P*-value = 9.8e-3, 1.50 ± 0.79; *P*-value < 2.2e-16, 3.70 ± 3.08 for the low-, moderate-, and high-transmission simulations, respectively) (Fig 1B).

As expected, the *var* genes measurement model, which accounts for potential sampling errors by sub-sampling the number of *var* genes per strain, reduced the number of available distinct *var* genes per host and increased inaccuracy (Figs 1A and 2). MOI estimates from simulated data can now differ from the true MOI by up to 2, 4, and 12 co-infections for the low-, moderate-, and high-transmission simulations, respectively. Overall, the distribution of the estimated MOI remained quite similar to that of the true MOI (*P*-values = 2.5e-10; average estimated MOI of 1.02 ± 0.14; *P*-value < 2.2e-16, 1.25 ± 0.48; *P*-value < 2.2e-16, 2.51 ± 1.75 for the low-, moderate-, and high-transmission simulations, respectively), even though it accentuated some of the small deviations we described in the absence of measurement error, namely more hosts with low MOI and fewer hosts with high MOI (Fig 1B).

As the *var*coding approach estimates MOI using individual host level information, subsampling the dataset, from 2000 to 500 or 200 individuals, did not reduce accuracy regardless of consideration of measurement error (S7 and S8 Figs). Sampling throughout the malaria season, instead of one time-point, did not influence the performance of the *var*coding approach (S9 and S10 Figs). Similarly, considering samples after a transient intervention or adding a background clearance of infections, due to processes that are not related to immune escape through the *var* gene memory, did not change the accuracy of the estimates (S9, S11, and S12 Figs).

The high-transmission setting simulations resulted in more accurate MOI estimates than the low- and moderate-transmission simulations (Fig 1A). This is consistent with the *var* repertoires for simulations under high-transmission settings exhibiting a lower average pairwise type sharing (PTS, see Materials and Methods) (4.7e-4 ± 1.8e-4), i.e., less overlap, than the *var* repertoires for simulations under low- and moderate-transmission settings (0.164 ± 0.043 and 0.049 ± 0.004, respectively) (Figs 4C, S15, and S16). The genetic structure of the parasite population can also be analyzed using networks whose nodes are *var* repertoires, and the weighted edges correspond to the degree of overlap between these repertoires (Fig 4A) [36]. Consistent with the average PTS, the similarity networks for simulations under high-transmission settings did not group the *var* repertoires into well-defined modules while the similarity networks for simulations under low- and moderate-transmission settings did. This finding was also captured using three-node motifs across the *var* repertoire similarity networks, which showed a lower proportion of reciprocal motifs (i.e., A ↔ B ↔ C ↔ A) for simulations under high-transmission settings (2.4% ± 6.4%) than for simulations under low- and moderate-transmission settings (11.8% ± 7.0% and 30.0% ± 3.2%, respectively) (Figs 4A, 4B and S8). Altogether, as simulated *P. falciparum* strains under high-transmission settings shared less similar *var* repertoires than those under low- or moderate-transmission settings, the *var*coding approach results in more accurate MOI estimates under high-transmission settings than under the lower transmission ones (Figs 2A, 4, and S15).

## Comparison of THE REAL McCOIL and the *var*coding approaches

As described above, each method performs better under specific conditions, such as transmission setting and sampling size. However, for any given simulation, THE REAL McCOIL approach never reached the level of accuracy of *var*coding (Fig 1). First, for simulations under low- and moderate-transmission settings, THE REAL McCOIL approach could generate highly inaccurate MAF, due to the small proportion of infected hosts sampled from the participants, which can result in more inaccurate MOI estimates than those generated with

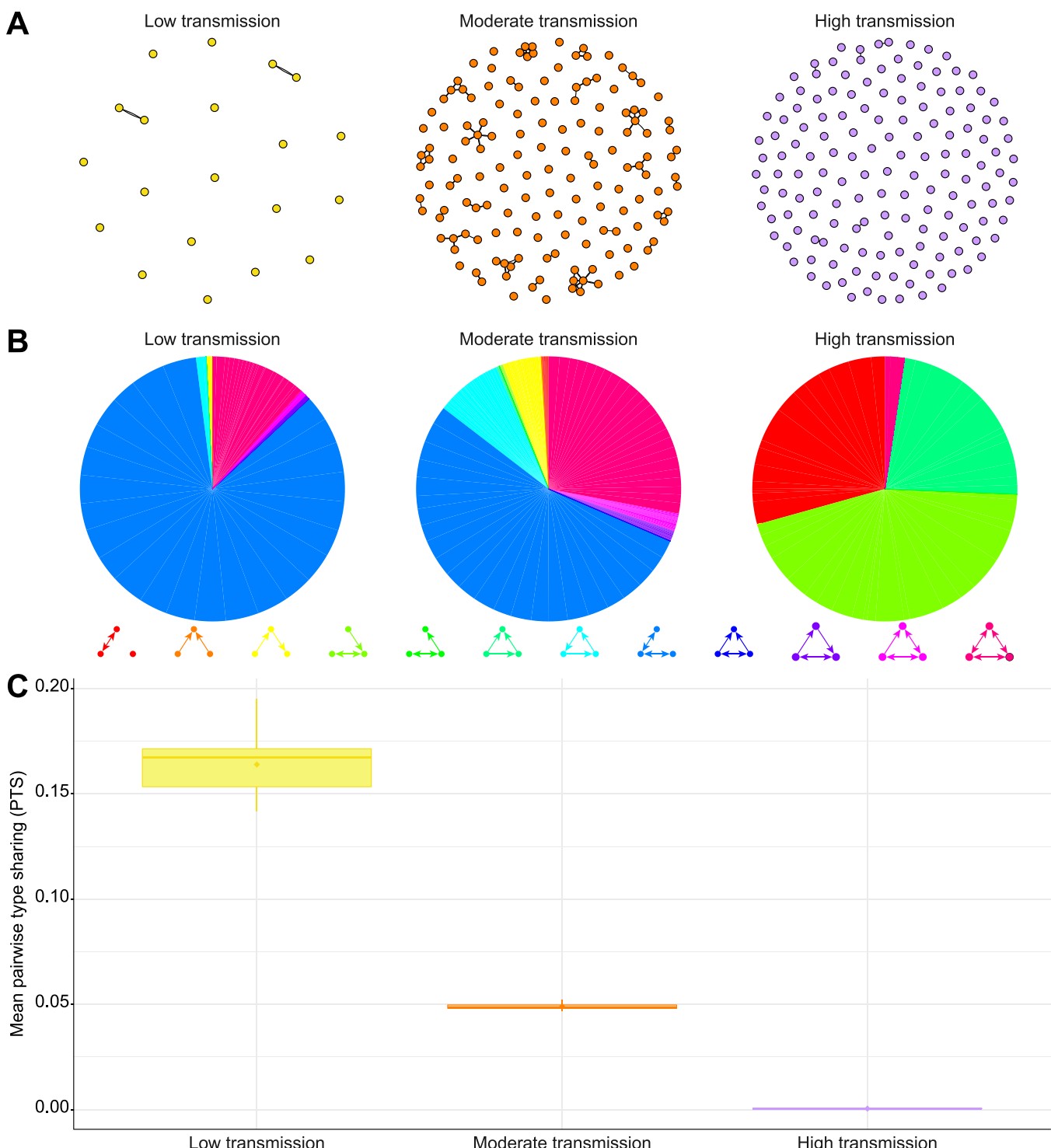

**Fig 4. Population structure using network properties.** Comparisons of repertoire similarity networks of 150 randomly sampled parasite *var* repertoires generated from a one-time point under low-, moderate-, and high-transmission settings (i.e., one replicate of runs 1, 25, and 49, respectively; S1 and S2 Tables). Only the top 1% of edges are drawn and used in the analysis. **A)** Similarity networks where nodes are *var* repertoires, weighted edges encode the degree of overlap between the *var* genes contained in these repertoires, and the direction of an edge indicates the asymmetric competition between repertoires. **B)** Distributions of the average proportion of occurrences of three-node graph motifs across the repertoire similarity networks. **C)** Distribution of the mean pairwise type sharing (PTS) between *var* repertoires. For each category, the horizontal central solid line represents the median, the diamond represents the mean, the box represents the interquartile range (IQR) from the 25th to 75th centiles, the whiskers indicate the most extreme data point, which is no more than 1.5 times the interquartile range from the box, and the dots show the outliers, i.e., the points beyond the whiskers.

*var*coding. Second, under high-transmission settings, THE REAL McCOIL approach showed a combination of MOI overestimates and underestimates for true MOI under or above ~14 co-infections, respectively (Fig 1). This introduces biases in opposite directions, which can compensate to some extent and artificially provide a reasonable overall population distribution. In contrast, the *var*coding approach showed consistent increasing underestimation of the MOI for increasing true MOI as expected from an increasing *var* repertoire overlap between strains, Third, for a similar sample size (i.e., 2000, 500, or 200 individuals), the *var*coding approach always provided more accurate MOI estimates than THE REAL McCOIL approach (S7 and S8 Figs). This was especially observed for the smaller sample sizes (i.e., 200 samples), which are quite commonly used for malaria surveillance. On the one hand, THE REAL McCOIL, which relies on information at the population level, could not provide MOI estimates due to the limited number of SNP loci that could be considered in the analysis or could provide less accurate MAF and MOI estimates. On the other hand, the *var*coding, which only relies on the information at the individual level to estimate MOI, consistently provided comparable MOI estimates independently of the sample size. In summary, the accuracy of the estimated MOI was dependent on the transmission setting, the approach used to characterize the multiplicity of malaria parasite infection, and the sample size.

## Discussion

When *P. falciparum* transmission is high, which is common within malaria-endemic regions, simulations showed that THE REAL McCOIL approach provided less robust MOI estimation than the *var*coding approach. The former approach tends to overestimate the MOI for hosts with low and moderate true MOI (under ~14 co-infections) and to underestimate the MOI for hosts with high true MOI (above ~14 co-infections). The high proportion of DACs and low proportion of single major and minor allele calls in a host SNP haplotype (barcode) seemed to be the origin of this inaccuracy in the high-transmission simulations. It is interesting to note that the combination of underestimated and overestimated MOI values allowed THE REAL McCOIL approach to generate a fairly accurate average estimated MOI but for the wrong reasons. Therefore, caution should be taken with using this approach when malaria transmission is moderate to high. In particular, considerable overestimates in the population result in a secondary peak in the distribution. In contrast, the low PTS values at high transmission due to the high diversity of *var* genes and selection for reduced repertoire overlap enabled accurate MOI estimates via *var*coding by amplifying, pooling, sequencing, and counting the number of DBLα types in a host. Despite harboring a high proportion of distinct *var* genes, repertoires can still be partially overlapping, sharing similar *var* genes. This limited overlap can result in a reduced number of *var* genes being identified on the basis of their DBLα types, which leads to the potential underestimation of MOI. Because these regions of the parasite genome can be sometimes challenging to access, the resulting sampling errors can reduce the reliability of the methodology leading to consistent underestimation of MOI, as shown with the simulations that included a realistic measurement error based on empirical data. Simulations including a measurement error based on the distribution of the number of non-upsA DBLα *var* gene types per 3D7 laboratory isolate significantly improved the accuracy of the *var* coding MOI estimates, highlighting the importance of high-density isolates when estimating MOI (S17 Fig). Moreover, extensions of the initial simulations supported the robustness of estimation with this method. Namely, consideration of transient vs. stationary dynamics, with sampling following a temporary intervention, did not modify our results; neither did sampling throughout the transmission season, nor the extension of the model to consider additional processes of infection clearance, through a general, background, clearance of infection. Thus, the *var*coding

approach provides a cost-effective approach for evaluating MOI under high-transmission conditions, with an underestimation bias introduced however by measurement error from the sampling of the *var* genes. This error can be taken into account with a Bayesian extension of the estimation method itself currently underway.

In low or moderate malaria transmission regions, both THE REAL McCOIL and *var*coding approaches can provide reasonable MOI estimates. As bi-allelic SNP data can be relatively cheap and straightforward to obtain, THE REAL McCOIL method, which can be applied to any parasite isolates with multiclonal infections [21], appears cost-effective for evaluating MOI but could nevertheless introduce biases in determining malaria elimination status by underestimating the effectiveness of the interventions. However, the method required a minimum number of sampled hosts to reasonably estimate MOI, which is not the case for *var*coding. Caution should thus be taken when defining the most suitable sample size while keeping this method cost-effective. Given the high accuracy of the *var*coding approach when measurement error was not incorporated, future work will address correcting for *var* repertoire overlap within a single host to improve MOI estimation.

Consistently with Chang *et al*. 2017, although THE REAL McCOIL approach assumes that genotyped SNP loci do not exhibit significant linkage disequilibrium (LD), the simulations performed with linked SNP loci did not show less reliable MOI estimation. Linked SNP loci may require longer simulations than the ones considered here (with thousands of generations) to show substantial bias in estimates of MOI. The categorical method of THE REAL McCOIL approach was very sensitive to the parameter controlling the upper bound for MOI (maxCOI). While a maximum MOI of 20 was applied in the simulations, reducing this upper bound for the low- and moderate-transmission simulations to the highest true MOI values observed for these settings (to 4 and 9 for the low and moderate-transmission simulations, respectively) significantly improved the accuracy of the estimated MOI (S18 Fig). Defining the most suitable MOI upper bound from previous reports could thus be useful to provide more accurate future estimates when using this approach. However, this solution can be circular and therefore impractical.

THE REAL McCOIL approach provided highly accurate MAF estimates for low- transmission intensities and reasonably accurate ones in moderate- and high-transmission intensities. While MAF estimates were robust with as few as 24 SNPs, their accuracy was improved by increasing the number of SNPs genotyped. Most population genetic analyses of malaria parasites rely on monoclonal infections, which reduces the amount of data and produces MAF estimates that may not be representative and thus introduce potential biases. Therefore, despite often significantly overestimating the MOI, THE REAL McCOIL approach could also facilitate population genetic analyses of the malaria parasite by properly estimating the MAF and other related statistics, including the effective population size ($N_e$), the $F_{ST}$, and the $F_{WS}$ [38–40].

An additional source of measurement error, not considered by our simulations, concerns the difficulty of properly sampling all the strains simultaneously infecting a particular host due to low parasitemia, the parasite load in the host blood. Indeed, multiclonal infections can potentially result in a reduction of the parasitemia of particular strains [41]. Consequently, strains with low parasitemia could be highly diluted in clinical samples and thus have a lower probability of being properly sequenced and/or genotyped. Moreover, infections (monoclonal or multiclonal) with low total parasitemia could be missed by the PCR detection approach, which can identify infections with as low as one parasite per μL, and by the commonly used microscopy detection approach, which cannot detect infections with lower than 4–10 parasites per μL [42,43]. Synchronicity of clones in the 48-hour life cycle on alternate days also leads to underestimation of MOI unless repeat daily sampling [44] or every three days is done [45]. These issues could therefore contribute to a more substantial underestimation of the MOI than the ones highlighted in this study.

Another simplification in our ABM was the lack of SNP mutations. Evolution of the neutral part of the parasite genome may influence MOI estimation over long time scales, but should play a minor role for the shorter time periods relevant to epidemiology unless associated with selective sweeps. Finally, our model did not incorporate the different sequence groupings of *var* genes, which can be classified based on their chromosomal position and semi-conserved upstream promoter sequences (ups) into different groups, upsA and non-upsA [46–48]. Case-control studies have reported that while upsA *var* genes are preferentially expressed in children with cerebral and/or severe malaria, non-upsA *var* genes have been associated with asymptomatic infections and clinical cases of malaria [49–57]. This absence of *var* types in our ABM may explain the higher prevalence within the younger age class in the simulations (i.e., 0–5 years old), compared to an observed higher prevalence typically within the 6–10 and 11–20 years old age classes in the empirical data [58].

Our ABM involved another simplification for the sexual reproduction stage of the parasite which assumes that the source of the donor parasites is always a single host. As genetic diversity and transmission intensity are highly interconnected, we do not expect this implementation of the sexual reproduction stage of the parasite to influence the results for high-transmission settings, when *var* repertoire overlap is extremely limited. Moreover, as interrupted feeds, where a single genotype from the same mosquito is inoculated into two people, can occur but not frequently [59], we also do not expect our lack of consideration of this process to change our results for low- and moderate-transmission settings.

Relatedness among recombined strains from the same mosquito should influence MOI estimation. As relatedness increases, it should be more difficult to differentiate separate strains that are related. Although our model does not explicitly represent mosquitoes, it naturally generates strains with different levels of relatedness from the complex interaction between meiotic recombination, transmission intensity, MOI distribution, and host immune selection. However, we do not expect relatedness to be an issue for our method at low-transmission, as recombination does not play an important role and MOI is sufficiently low that transmission events mostly concern donor infections with a single host. At high transmission, we also do not expect relatedness arising from recombination to be an important source of error, given the realistic implementation of recombination (as described in the Materials and Methods).

Although MOI is a useful epidemiological marker to evaluate the efficacy of malaria intervention efforts, properly characterizing multiclonal infections remains a challenge, especially for high transmission. This work demonstrates that THE REAL McCOIL and the *var*coding approaches provide complementary methodologies to determine MOI across distinct transmission settings. In particular, the high diversity of the *var* gene family and low overlap of *var* gene repertoires between parasites, especially under high-transmission intensity, allows robust MOI estimation with the *var*coding method, despite a tendency for underestimation originating mainly from sampling error. Reliance of THE REAL McCOIL on bi-allelic neutral SNPs limits application at high transmission, with the introduction of a secondary peak in the tail of the population distribution, considerable over-estimates of individual MOI, and opposite signs in the deviations, for both under- and over-estimates in different ranges of true values. The method provides reasonable estimates across low- and moderate-transmission settings where the *var*coding approach could be limited by partially overlapping *var* repertoires. Considering local transmission intensity is thus highly recommended when defining the most suitable marker and/or MOI estimation approach to evaluate the impact of malaria control and elimination campaigns. The highly diverse multigene *var* family under immune selection provides a handle to complexity of infection at high transmission.

## Materials and methods

### Agent-Based Model (ABM)

Malaria transmission was modeled with an extended implementation of an agent-based, discrete-event, stochastic model in continuous time [36,60]. Here, we briefly describe the agent-based model (ABM) implemented in Julia (varmodel3) which is based on previous C++ implementations (i.e. varmodel and varmodel2) [36,60]. While the previous implementation of the stochastic ABM was adapted from the next-reaction method which optimizes the Gillespie first-reaction method, this implementation uses a simpler Gillespie algorithm [61,62]. The ABM tracks the infection history and immune memory of each host and its parameters and symbols are summarized in S1 Table. We modeled a local population of 10000 individuals, and a global *var* gene pool whose size acts as a proxy for regional parasite diversity. The simulations are initialized with 20 migrant infections from this regional pool to seed local population transmission and grow local gene diversity to a stationary equilibrium. Each migrant parasite genome consists of a specific combination (i.e. repertoire) of 45 *var* genes. The size of the repertoire was based on the median number of non-upsA DBLα sequences identified in our 3D7 laboratory isolate [22,23]. This grouping of *var* genes is based on their semi-conserved upstream promoter sequences (ups) (i.e. upsA and non-upsA (upsB and upsC)) [46–48]. Although each parasite carries both types of *var* genes in a fairly constant proportion [26,63], the MOI estimation method we consider here focuses on the non-upsA DBLα sequences as they were ~20X more diverse and less conserved among repertoires than the upsA DBLα sequences. Therefore, for simplicity purposes, our model considered only those types. Each *var* gene itself is represented as a linear combination of two epitopes, i.e. parts of the molecule that act as antigens and are targeted by the immune system [26,36,64]. The *var* genes in a repertoire are expressed sequentially and the infection ends when the whole repertoire is depleted. The duration of the active period of a *var* gene, and thus of the infection, is determined by the number of unseen epitopes. When a *var* gene is deactivated, the host adds the deactivated *var* gene epitopes to its immunity memory. Specific immunity toward a given epitope experiences a loss rate from host immunity memory, and re-exposure is therefore required to maintain it. The local population is open to immigration from the regional pool.

Our model extension allows us to keep track of the neutral part of each migrant parasite genome assembled by sampling one of the two possible alleles (labeled as 0 or 1) at each of a defined number of neutral bi-allelic SNPs (S1 Table). While the extended model can generate homogeneous initial SNP allele frequencies by sampling the migrant alleles with an identical probability from the regional pool (i.e., 0.5), it can also generate distinct initial SNP allele frequencies by sampling the migrant alleles from the regional pool with distinct initial probabilities that sum up to one (e.g., 0.2 and 0.8) and that are randomly and uniformly picked from a defined range (e.g., [0.1–0.9]; S1 Table).

Seasonality was implemented in the transmission rate parameter to represent monthly variability in mosquito bites [60,65]. The model does not explicitly incorporate mosquito vectors but considers instead an effective contact rate (hereafter, the transmission rate) which determines the times of local transmission events (exponentially distributed). At these times, a donor and a recipient host are selected randomly. We do not model oocyst tetrads directly, but we assume that in a transmission event the number of strains that survive in the blood stage of the recipient host equals the number of strains picked up from the donor host, say $N_s$. Because meiotic recombination happens within the mosquito during the sexual stage of the parasite, it is implemented in the model at the time of a transmission event. Specifically, the process of generating each daughter strain is as follows: 1) two parent strains are selected at random (these could be the same strain), 2) two recombined daughter strains are generated, and one is

randomly selected for transmission. Thus, strains selected during a transmission event have a probability $P_r = 1 - 1 / N_s$ to become a recombinant strain, because all the strains present in the mosquito are assumed to be of similar frequency and have an equal chance of forming zygotes with any strains including itself [36]. This process mimics how the oocysts are formed and sporozoites are produced in mosquitoes and is similar to what is considered in Wong *et al.* (2018) [66]. While the association of physical locations and major groups of *var* genes is established, orthologous gene pairs between two strains are often unknown or on different chromosomes. Therefore, to generate a recombinant *var* repertoire, a random set of *var* genes is sampled from a pool containing the two sets of *var* genes from the original genomes. As physical locations of *var* genes can be mobile because mitotic recombination or gene conversion occur between *var* genes at different physical locations constantly, this assumption is a reasonable simplification of the meiotic recombination process. Similarly, to generate the neutral part of a recombinant parasite, a random allele is sampled for each bi-allelic SNP. Moreover, to allow for linkage disequilibrium (LD) across the neutral part of the genome, neutral bi-allelic SNPs can be non-randomly associated and co-segregate as defined in a matrix of LD coefficients indicating the probability that pairs of linked SNPs will co-segregate during the meiotic recombination (S2 Table).

## Experimental design

We explored how distinct transmission settings influence MOI estimation with the two different approaches. Specifically, we compared three transmission intensities corresponding to "low" (prevalence of 1–10%), "moderate" (prevalence of 10–35%), and "high" (prevalence ≥35%), implemented with different transmission rates (5.0e-05, 7.5e-05, and 1.0e-04, respectively), and initial gene pool sizes (500, 2000, and 10000, respectively) (S1 and S2 Tables) [67]. As the sensitivity of the SNP-based methods increases with the number of SNP loci, and as Chang *et al.* (2017) retained 105 SNP loci to test THE REAL McCOIL approach, we performed these simulations using 24, 48, 96, and 105 SNP loci for the three transmission settings (S1 and S2 Tables) [21,24,25]. As THE REAL McCOIL approach assumes that distinct parasite lineages in multiclonal infections are unrelated and that genotyped SNP loci do not exhibit significant LD, we performed the simulations with homogenous initial SNP allelic frequencies and with unlinked bi-allelic SNP loci (S1 and S2 Tables). However, as allelic frequencies can be heterogeneous in space and time, we also performed simulations with distinct initial allelic frequencies, and with 8% and 16% of linked SNP loci clustered into one or two groups, respectively. To avoid simulations that may contain too many fixed SNP loci, the distinct initial allele frequencies were randomly and uniformly picked from a defined range only allowing frequencies greater than 10% or lower than 90% (S1 Table). However, while all loci have an initial MAF ≥ 10%, their SNP allele frequencies can change over time as a function of the transmission dynamics of *P. falciparum*. This design results in 72 distinct combinations of parameters (i.e., runs) and we ran 10 replicates per combination with a maximum MOI of 20 (S1 and S2 Tables). Simulations were run for 85 years to get beyond the initial transient dynamics in which *var* gene diversity and parasite population structure are established. For each simulation, we calculated the epidemiological summary statistics, including the number of hosts, the prevalence, and the entomological inoculation rate (EIR). In addition, 2000 individuals were randomly sampled to analyze the true MOI and the parasite genetic and allelic diversity patterns. The simulated data were collected during the last year at 300 days (i.e., November), corresponding to the end of the wet season (high-transmission season) in the Bongo District, a malaria-endemic area of northern Ghana. Details on the area and population have been previously described [23,58].

## MOI estimation

While the "true" MOI per host was directly extracted from the simulations, the estimated MOI was obtained for each host using the two distinct approaches. First, the MOI per host was estimated from the simulated neutral SNP data using the mean MOI provided by THE REAL McCOIL approach v.2 [21]. We performed the categorical method of THE REAL McCOIL with a minor allele frequency (MAF) of 10% for a SNP to be considered and an upper bound of 20 for MOI, keeping all other parameters to their default values (a burn-in period of $10^3$ iterations, a total of $10^4$ Markov chain Monte Carlo (MCMC) iterations, a minimum number of 20 genotypes for an individual to be considered, a minimum number of 20 samples for a SNP to be considered, an initial MOI of 15, and an initial probability of calling single allele loci double allele loci and of calling double allele loci single allele loci of 0.05 which were estimated with MOI and the allele frequencies) [21,68]. Loci with MAF < 10% were removed from the analyses as they indicated loci that are not representative, that alleles were moving towards fixation in the population, and that would not be informative to differentiate isolates from each other in the population. Second, the MOI per host was also estimated from the simulated *var* genes data by counting the total number of distinct *var* genes within each host and by dividing it by the size of one repertoire, here 45 as estimated from control data using repeat samplings of the *var* genes of 3D7 with the *var*coding protocol [23].

To account for measurement error in both approaches, a measurement model was implemented. First, to account for potential SNP genotyping failures, we applied a measurement model that randomly replaces the host genotypes with missing data, reducing the number of available data for THE REAL McCOIL approach (Fig 2). This replacement was implemented by using the distribution of the proportion of missing genotypes per monoclonal infections from a panel of 24 bi-allelic SNP loci which was previously obtained during one cross-sectional survey in 2015 in the Bongo District in Ghana at the end of the wet season [24,69] (Fig 1C). For each host, some SNP loci were thus replaced with missing genotypes according to a weight reflecting the proportion of missing genotype counts density function. Second, to account for *var* gene potential sampling errors, we applied a measurement model that sub-samples the number of *var* genes per strain, resulting in a reduction of the total number of *var* genes per host (Fig 2). This sub-sampling was implemented by exploring the distribution of the number of non-upsA DBLα *var* gene types per monoclonal infection for which molecular sequences were previously obtained during six cross-sectional surveys between 2012 and 2016 in the Bongo District in Ghana at the end of the wet season [22,23,36,60] (Fig 1D). For each strain, the number of *var* genes was sub-sampled according to a weight reflecting the *var* gene counts density function.

MOI estimations were carried out with and without measurement error. To better reflect what is typically done for malaria surveillance, we also estimated the MOI after subsampling the simulated dataset (from 2000 to 500 or 200 individuals) and after sampling a total of 2000 individuals collected through five time-points during the malaria season (180, 210, 240, 270, and 300 days; 400 sampled individuals per time-point) instead of one single time-point. Moreover, as these approaches are used to estimate MOI in regions where transmission intensity has recently changed (e.g., due to a successful intervention or a rebound post-intervention [70]), we explored how each approach performs after we applied a temporary intervention reducing the biting rate by 50% during the two years preceding the sampling year (specifically, between years 83 and 84). Finally, to consider the possibility that infections could resolve randomly before exhaustion of the *var* gene repertoire (e.g., due to other mechanisms of infection control), we also explored how background clearance due to processes not explicitly modeled impacts the performance of the estimated MOI ("background_clearance_rate" = 0.001; S1

Table). The impact of a background clearance process and recent change in transmission intensity were investigated in parallel by performing simulations having 48 unlinked SNP loci with homogenous initial allelic frequencies for the three transmission settings, resulting in three distinct combinations with 10 replicates per combination. T-tests were used to compare true and estimated MOI distributions. All t-test comparisons were considered statistically significant when $P$-value $\leq 0.05$.

## Repertoire similarity networks

To evaluate the similarity of parasites in the population, pairwise type sharing (PTS) was calculated between all repertoire pairs (regardless of the host in which they are encountered) as $PTS_{ij} = 2n_{ij} / (n_i + n_j)$, where $n_i$ and $n_j$ are the number of unique *var* genes within each repertoire $i$ and $j$ and $n_{ij}$ is the total number of *var* genes shared between repertoires $i$ and $j$ [28]. In addition, the genetic structure of the *P. falciparum* population was also analyzed using similarity networks based on *var* composition. Similarity networks were built in which nodes are *var* repertoires, weighted edges encode the degree of overlap between the *var* genes contained in these repertoires, and the direction of an edge indicates the asymmetric competition between repertoires, i.e., whether one repertoire can outcompete the other [36,71]. To introduce directional edges, we calculated the genetic similarity of repertoire $i$ to repertoire $j$ as $S_{ij} = (N_i \cap N_j) / N_i$, where $N_i$ and $N_j$ are the number of unique *var* genes in repertoires $i$ and $j$, respectively. To focus on the *var* repertoires with the strongest overlap, only the top 1% of edges are drawn and used in network analysis.

## Supporting information

**S1 Fig. Host age distribution, and number of sampled individuals and hosts per age class.** For each category, the horizontal central solid line represents the median, the diamond represents the mean, the box represents the interquartile range (IQR) from the 25th to 75th centiles, the whiskers indicate the most extreme data point which is no more than 1.5 times the interquartile range from the box, and the dots show the outliers, i.e. the points beyond the whiskers. **A)** Age distribution of the sampled individuals. **B)** Number of sampled individuals per age class. **C)** Number of sampled hosts per age class. Upper (yellow), middle (orange), and lower (purple) panels correspond to simulations under low-, moderate-, and high-transmission settings, respectively (S1 and S2 Tables). Values were split into five age classes, i.e. 0–5, 6–10, 11–20, 21–39, and $\geq 40$ years.
(EPS)

**S2 Fig. Prevalence and entomological inoculation rate (EIR) per transmission intensity.** For each category, the horizontal central solid line represents the median, the diamond represents the mean, the box represents the interquartile range (IQR) from the 25th to 75th centiles, the whiskers indicate the most extreme data point which is no more than 1.5 times the interquartile range from the box, and the dots show the outliers, i.e. the points beyond the whiskers. **A)** Prevalence; **B)** EIR. Statistics calculated for simulations under low-, moderate-, and high-transmission settings are indicated in yellow, orange, and purple, respectively (S1 and S2 Tables).
(EPS)

**S3 Fig. Proportion of SNP calls (genotypes) per host SNP haplotype and per transmission intensity. A)** Single major allele calls; **B)** Double allele calls (DACs); **C)** Single minor allele calls; **D)** Missing allele calls. Each SNP call proportion was calculated using the low-, moderate-, and high-transmission setting simulations (S1 and S2 Tables). The column panels show

the proportions for specific true MOI values. The dark and light green colors indicate the proportion of calls made without and with a measurement model, respectively (Fig 2). For each category, the horizontal central solid line represents the median, the diamond represents the mean, the box represents the interquartile range (IQR) from the 25th to 75th centiles, the whiskers indicate the most extreme data point, which is no more than 1.5 times the interquartile range from the box, and the dots show the outliers, i.e., the points beyond the whiskers. (EPS)

**S4 Fig. THE REAL McCOIL 95% credible intervals of the estimated MOI.** For each category, the horizontal central solid line represents the median, the diamond represents the mean, the box represents the interquartile range (IQR) from the 25th to 75th centiles, the whiskers indicate the most extreme data point, which is no more than 1.5 times the interquartile range from the box, and the dots show the outliers, i.e., the points beyond the whiskers. Statistics calculated for simulations under low-, moderate-, and high-transmission settings are indicated in yellow, orange, and purple, respectively (S1 and S2 Tables). (EPS)

**S5 Fig. Initial number of SNPs and accuracy of the multiplicity of infection (MOI) estimates determined with THE REAL McCOIL approach.** The accuracy of MOI estimates is defined as the differences between estimated and true MOI per host. While null values highlight accurate MOI estimates (indicated by a dashed black horizontal line), the positive and negative values highlight over- and under-estimation, respectively. The dark and light green colors indicate respectively MOI estimations made without and with a measurement model (Fig 2). For each category, the horizontal central solid line represents the median, the diamond represents the mean, the box represents the interquartile range (IQR) from the 25th to 75th centiles, the whiskers indicate the most extreme data point which is no more than 1.5 times the interquartile range from the box, and the dots show the outliers, i.e. the points beyond the whiskers. **A)** Accuracy of MOI estimates per true MOI. **B)** Accuracy of MOI estimates per transmission intensity (S1 and S2 Tables). (EPS)

**S6 Fig. SNP properties and accuracy of the multiplicity of infection (MOI) estimates determined with THE REAL McCOIL approach.** The accuracy of MOI estimates is defined as the differences between estimated and true MOI per host. While null values highlight accurate MOI estimates (indicated by a dashed black horizontal line), the positive and negative values highlight over- and under-estimation, respectively. For each category, the horizontal central solid line represents the median, the diamond represents the mean, the box represents the interquartile range (IQR) from the 25th to 75th centiles, the whiskers indicate the most extreme data point which is no more than 1.5 times the interquartile range from the box, and the dots show the outliers, i.e. the points beyond the whiskers. The dark and light green colors indicate respectively MOI estimations made without and with a measurement model (Fig 2). (EPS)

**S7 Fig. Reliability of the multiplicity of infection (MOI) estimations when subsampling 25% of the sampled individuals (i.e. 500 individuals).** For each category, the horizontal central solid line represents the median, the diamond represents the mean, the box represents the interquartile range (IQR) from the 25th to 75th centiles, the whiskers indicate the most extreme data point which is no more than 1.5 times the interquartile range from the box, and the dots show the outliers, i.e. the points beyond the whiskers. The upper, middle, and lower row panels correspond to simulations under low-, moderate-, and high-transmission settings, respectively (S1 and S2 Tables). **A)** Accuracy of MOI estimates, defined as the difference

between estimated and true MOI per host. While null values highlight accurate MOI estimates (indicated by a dashed black horizontal line), the positive and negative values highlight over- and under-estimation, respectively. Estimates with the neutral SNP-based approach (THE REAL McCOIL) are indicated in green, and those with the *var* gene-based approach (*var*coding) are indicated in blue. The dark and light green or blue colors indicate respectively MOI estimations made without and with a measurement model (Fig 2). The column panels show differences for specific true MOI values. **B)** Population distribution of the estimated and true MOI per host from the simulated "true" values and those estimated with the methods indicated by the colors similar to panel A.
(EPS)

**S8 Fig. Reliability of the multiplicity of infection (MOI) estimations when subsampling 10% of the sampled individuals (i.e. 200 individuals).** For each category, the horizontal central solid line represents the median, the diamond represents the mean, the box represents the interquartile range (IQR) from the 25th to 75th centiles, the whiskers indicate the most extreme data point which is no more than 1.5 times the interquartile range from the box, and the dots show the outliers, i.e. the points beyond the whiskers. The upper, middle, and lower row panels correspond to simulations under low-, moderate-, and high-transmission settings, respectively (S1 and S2 Tables). **A)** Accuracy of MOI estimates, defined as the difference between estimated and true MOI per host. While null values highlight accurate MOI estimates (indicated by a dashed black horizontal line), the positive and negative values highlight over- and under-estimation, respectively. Estimates with the neutral SNP-based approach (THE REAL McCOIL) are indicated in green, and those with the *var* gene-based approach (*var*coding) are indicated in blue. The dark and light green or blue colors indicate respectively MOI estimations made without and with a measurement model (Fig 2). The column panels show differences for specific true MOI values. **B)** Population distribution of the estimated and true MOI per host from the simulated "true" values and those estimated with the methods indicated by the colors similar to panel A.
(EPS)

**S9 Fig. Reliability of the multiplicity of infection (MOI) estimations for only one combination of parameters (i.e. run) per transmission intensity (i.e. run 12, 36, and 60 for the low-, moderate-, and high-transmission intensities, respectively).** For each category, the horizontal central solid line represents the median, the diamond represents the mean, the box represents the interquartile range (IQR) from the 25th to 75th centiles, the whiskers indicate the most extreme data point which is no more than 1.5 times the interquartile range from the box, and the dots show the outliers, i.e. the points beyond the whiskers. The upper, middle, and lower row panels correspond to simulations under low-, moderate-, and high-transmission settings, respectively (S1 and S2 Tables). **A)** Accuracy of MOI estimates, defined as the difference between estimated and true MOI per host. While null values highlight accurate MOI estimates (indicated by a dashed black horizontal line), the positive and negative values highlight over- and under-estimation, respectively. Estimates with the neutral SNP-based approach (THE REAL McCOIL) are indicated in green, and those with the *var* gene-based approach (*var*coding) are indicated in blue. The dark and light green or blue colors indicate respectively MOI estimations made without and with a measurement model (Fig 2). The column panels show differences for specific true MOI values. **B)** Population distribution of the estimated and true MOI per host from the simulated "true" values and those estimated with the methods indicated by the colors similar to panel A.
(EPS)

**S10 Fig. Reliability of the multiplicity of infection (MOI) estimations when hosts are sampled through several time-points during the malaria season.** Only one combination of parameters (i.e. run) per transmission intensity (i.e. run 12, 36, and 60 for the low-, moderate-, and high-transmission intensities, respectively) is illustrated. For each category, the horizontal central solid line represents the median, the diamond represents the mean, the box represents the interquartile range (IQR) from the 25th to 75th centiles, the whiskers indicate the most extreme data point which is no more than 1.5 times the interquartile range from the box, and the dots show the outliers, i.e. the points beyond the whiskers. The upper, middle, and lower row panels correspond to simulations under low-, moderate-, and high-transmission settings, respectively (S1 and S2 Tables). **A)** Accuracy of MOI estimates, defined as the difference between estimated and true MOI per host. While null values highlight accurate MOI estimates (indicated by a dashed black horizontal line), the positive and negative values highlight over- and under-estimation, respectively. Estimates with the neutral SNP-based approach (THE REAL McCOIL) are indicated in green, and those with the *var* gene-based approach (*var*coding) are indicated in blue. The dark and light green or blue colors indicate respectively MOI estimations made without and with a measurement model (Fig 2). The column panels show differences for specific true MOI values. **B)** Population distribution of the estimated and true MOI per host from the simulated "true" values and those estimated with the methods indicated by the colors similar to panel A.
(EPS)

**S11 Fig. Reliability of the multiplicity of infection (MOI) estimations when infection can be cleared by processes not explicitly modelled.** Only one combination of parameters (i.e. run) per transmission intensity (i.e. run 12, 36, and 60 for the low-, moderate-, and high-transmission intensities, respectively) is illustrated. For each category, the horizontal central solid line represents the median, the diamond represents the mean, the box represents the interquartile range (IQR) from the 25th to 75th centiles, the whiskers indicate the most extreme data point which is no more than 1.5 times the interquartile range from the box, and the dots show the outliers, i.e. the points beyond the whiskers. The upper, middle, and lower row panels correspond to simulations under low-, moderate-, and high-transmission settings, respectively (S1 and S2 Tables). **A)** Accuracy of MOI estimates, defined as the difference between estimated and true MOI per host. While null values highlight accurate MOI estimates (indicated by a dashed black horizontal line), the positive and negative values highlight over- and under-estimation, respectively. Estimates with the neutral SNP-based approach (THE REAL McCOIL) are indicated in green, and those with the *var* gene-based approach (*var*coding) are indicated in blue. The dark and light green or blue colors indicate respectively MOI estimations made without and with a measurement model (Fig 2). The column panels show differences for specific true MOI values. **B)** Population distribution of the estimated and true MOI per host from the simulated "true" values and those estimated with the methods indicated by the colors similar to panel A.
(EPS)

**S12 Fig. Reliability of the multiplicity of infection (MOI) estimations when intervention changed the recent transmission intensities.** Only one combination of parameters (i.e. run) per transmission intensity (i.e. run 12, 36, and 60 for the low-, moderate-, and high-transmission intensities, respectively) is illustrated. For each category, the horizontal central solid line represents the median, the diamond represents the mean, the box represents the interquartile range (IQR) from the 25th to 75th centiles, the whiskers indicate the most extreme data point which is no more than 1.5 times the interquartile range from the box, and the dots show the outliers, i.e. the points beyond the whiskers. The upper, middle, and lower row panels

correspond to simulations under low-, moderate-, and high-transmission settings, respectively (S1 and S2 Tables). **A)** Accuracy of MOI estimates, defined as the difference between estimated and true MOI per host. While null values highlight accurate MOI estimates (indicated by a dashed black horizontal line), the positive and negative values highlight over- and under-estimation, respectively. Estimates with the neutral SNP-based approach (THE REAL McCOIL) are indicated in green, and those with the *var* gene-based approach (*var*coding) are indicated in blue. The dark and light green or blue colors indicate respectively MOI estimations made without and with a measurement model (Fig 2). The column panels show differences for specific true MOI values. **B)** Population distribution of the estimated and true MOI per host from the simulated "true" values and those estimated with the methods indicated by the colors similar to panel A.
(EPS)

**S13 Fig. Accuracy of the minor allele frequency (MAF) estimates per locus determined with THE REAL McCOIL approach.** The accuracy of MAF estimates per locus is defined as the differences between estimated and true MAF per locus. While null values highlight accurate MAF estimates per locus (indicated by a dashed black horizontal line), the positive and negative values highlight over- and under-estimation, respectively. For each category, the horizontal central solid line represents the median, the diamond represents the mean, the box represents the interquartile range (IQR) from the 25th to 75th centiles, the whiskers indicate the most extreme data point which is no more than 1.5 times the interquartile range from the box, and the dots show the outliers, i.e. the points beyond the whiskers. The dark and light green colors indicate respectively MAF estimations made without and with a measurement model (Fig 2). Upper, middle, and lower panels correspond to simulations under low-, moderate-, and high-transmission settings, respectively (S1 and S2 Tables).
(EPS)

**S14 Fig. Accuracy of MOI estimates, defined as the differences between estimated and true MOI per host, when only keeping simulations with the lowest estimated MAF inaccuracy per simulation (inaccuracy < 0.39 = 1st quantile), defined as the sum of the absolute differences between estimated MAF using THE REAL McCOIL and true MAF per SNP locus.** For each category, the horizontal central solid line represents the median, the diamond represents the mean, the box represents the interquartile range (IQR) from the 25th to 75th centiles, the whiskers indicate the most extreme data point which is no more than 1.5 times the interquartile range from the box, and the dots show the outliers, i.e. the points beyond the whiskers. The upper, middle, and lower row panels correspond to simulations under low-, moderate-, and high-transmission settings, respectively (S1 and S2 Tables). While null values highlight accurate MOI estimates (indicated by a dashed black horizontal line), the positive and negative values highlight over- and under-estimation, respectively. Estimates with the neutral SNP-based approach (THE REAL McCOIL) are indicated in green, and those with the *var* gene-based approach (*var*coding) are indicated in blue. The dark and light blue or green colors indicate respectively MOI estimates made without and with a measurement model (Fig 2). The column panels show differences for specific true MOI values.
(EPS)

**S15 Fig. Population structure using repertoire similarity network properties.** Comparisons of repertoire similarity networks of 150 randomly sampled parasite *var* repertoires generated from a one-time point under low, moderate, and high-transmission settings (S1 and S2 Tables). Only the top 1% of edges are drawn and used in the analysis. The upper panel shows the distribution of the mean pairwise type sharing (PTS) per run. For each category, the

horizontal central solid line represents the median, the diamond represents the mean, the box represents the interquartile range (IQR) from the 25th to 75th centiles, the whiskers indicate the most extreme data point which is no more than 1.5 times the interquartile range from the box, and the dots show the outliers, i.e. the points beyond the whiskers. The lower panel shows the distributions of the proportion of occurrences of three-node graph motifs across the repertoire similarity networks.
(EPS)

**S16 Fig. Pairwise type sharing (PTS).** For each category, the horizontal central solid line represents the median, the diamond represents the mean, the box represents the interquartile range (IQR) from the 25th to 75th centiles, and the whiskers indicate the most extreme data point. **A)** Distribution of the PTS per transmission intensity. **B)** Distribution of the PTS per run.
(EPS)

**S17 Fig. Reliability of the multiplicity of infection (MOI) estimations when simulations include a measurement error based on the distribution of the number of non-upsA DBLα *var* gene types per 3D7 laboratory isolates for the *var* coding approach.** For each category, the horizontal central solid line represents the median, the diamond represents the mean, the box represents the interquartile range (IQR) from the 25th to 75th centiles, the whiskers indicate the most extreme data point which is no more than 1.5 times the interquartile range from the box, and the dots show the outliers, i.e. the points beyond the whiskers. The upper, middle, and lower row panels correspond to simulations under low-, moderate-, and high-transmission settings, respectively (S1 and S2 Tables). **A)** Accuracy of MOI estimates, defined as the differences between estimated and true MOI per host. While null values highlight accurate MOI estimates (indicated by a dashed black horizontal line), the positive and negative values highlight over- and under-estimation, respectively. Estimates with the neutral SNP-based approach (THE REAL McCOIL) are indicated in green, and those with the *var* gene-based approach (*var*coding) are indicated in blue. The dark and light blue or green colors indicate respectively MOI estimates made without and with a measurement model (Fig 2). The column panels show differences for specific true MOI values. **B)** Population distribution of the estimated and true MOI per host from the simulated "true" values and those estimated with the methods indicated by the colors similar to panel A.
(EPS)

**S18 Fig. Reliability of the multiplicity of infection (MOI) estimations when THE REAL McCOIL approach using an upper bound for MOI of 4, 9, and 20 for the low-, moderate-, and high-transmission simulations, respectively.** For each category, the horizontal central solid line represents the median, the diamond represents the mean, the box represents the interquartile range (IQR) from the 25th to 75th centiles, the whiskers indicate the most extreme data point which is no more than 1.5 times the interquartile range from the box, and the dots show the outliers, i.e. the points beyond the whiskers. The upper, middle, and lower row panels correspond to simulations under low-, moderate-, and high-transmission settings, respectively (S1 and S2 Tables). **A)** Accuracy of MOI estimates, defined as the differences between estimated and true MOI per host. While null values highlight accurate MOI estimates (indicated by a dashed black horizontal line), the positive and negative values highlight over- and under-estimation, respectively. The estimated MOI using the *var* genes based approach (i.e. *var* coding) are indicated in blue, and the estimated MOI using the neutral SNPs based approach (i.e. THE REAL McCOIL) are indicated in green. The dark and light blue or green colors indicate respectively MOI estimates made without and with a measurement model (Fig

2). The column panels show differences for specific true MOI values. **B)** Population distribution of the estimated and true MOI per host from the simulated "true" values and those estimated with the methods indicated by the colors similar to panel A.
(EPS)

**S1 Table. Epidemiological and genetic parameters used in the stochastic simulations.**
(XLSX)

**S2 Table. Epidemiological and genetic distinct parameters per run.**
(XLSX)

**S3 Table. Pearson correlation coefficients between the inaccuracy of the minor allele frequency (MAF) per locus estimated with THE REAL McCOIL approach (defined as the absolute differences between estimated and true MAF per locus), and the locus properties.**
(XLSX)

## Acknowledgments

We thank the participants, communities, and the Ghana Health Service in Bongo District, Ghana, for their willingness to participate in this study. We would like to acknowledge the programming assistance of Edward B. Baskerville. We appreciate the support of the Research Computing Center at the University of Chicago through the computational resources of the Midway cluster.

## Author Contributions

**Conceptualization:** Frédéric Labbé, Karen P. Day, Mercedes Pascual.

**Data curation:** Frédéric Labbé.

**Formal analysis:** Frédéric Labbé.

**Funding acquisition:** Karen P. Day, Mercedes Pascual.

**Investigation:** Frédéric Labbé.

**Methodology:** Frédéric Labbé, Qixin He, Kathryn E. Tiedje, Dionne C. Argyropoulos, Mun Hua Tan.

**Project administration:** Karen P. Day, Mercedes Pascual.

**Resources:** Anita Ghansah, Mercedes Pascual.

**Software:** Frédéric Labbé, Qixin He.

**Supervision:** Karen P. Day, Mercedes Pascual.

**Validation:** Frédéric Labbé, Qixin He, Mercedes Pascual.

**Visualization:** Frédéric Labbé.

**Writing – original draft:** Frédéric Labbé, Qixin He, Qi Zhan, Kathryn E. Tiedje, Dionne C. Argyropoulos, Mun Hua Tan, Anita Ghansah, Karen P. Day, Mercedes Pascual.

**Writing – review & editing:** Frédéric Labbé, Qixin He, Qi Zhan, Kathryn E. Tiedje, Dionne C. Argyropoulos, Mun Hua Tan, Anita Ghansah, Karen P. Day, Mercedes Pascual.

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
