## [Decision Letter · Decision Letter 0]

23 Aug 2022

Dear Dr. Pascual,

Thank you very much for submitting your manuscript "Neutral vs. non-neutral genetic footprints of Plasmodium falciparum multiclonal infections" for consideration at PLOS Computational Biology.

As with all papers reviewed by the journal, your manuscript was reviewed by members of the editorial board and by several independent reviewers. While the reviewers found the work of interest to the field, that raised some major limitations of the manuscript and analysis in its current form. In light of the reviews (below this email), we would like to invite the resubmission of a significantly-revised version that takes into account the reviewers' and editors comments.

We cannot make any decision about publication until we have seen the revised manuscript and your response to the reviewers' comments. Your revised manuscript is also likely to be sent to reviewers for further evaluation.

Sincerely,

David S. Khoury

Academic Editor

PLOS Computational Biology

Virginia Pitzer

Deputy Editor-in-Chief

PLOS Computational Biology

**Editors Comments to the Authors:**

The authors present an interesting and relevant assessment of two methods for estimating the multiplicity of infection from population level parasite genetic data. This work essentially involves using an existing agent based model of malaria transmission, with extensions to monitor genetic features of the transmission, and applying the two procedures to the simulated data. The essential conclusions are that both methods have various levels of success and failure at different transmission intensities, but the authors better agreement of the mean MOIs estimated using the varcoding approach (compared with the true MOIs) than THE REAL McCOIL approach:

True mean MOIs: 1.08 ± 0.30, 1.61 ± 0.95, and 3.84 ± 3.31

THE REAL McCOIL: 1.27 ± 1.60, 1.93 ± 1.93, and 4.80 ± 4.60

Varcoding: 1.08 ± 0.29; 1.60 ± 0.93; and 3.74 ± 3.14

While this study appears interesting and the results sensibly analysed, the major conclusions of the study may be highly dependent on the particular transmission model used to simulate the data, and in particular, the mechanisms of genetic diversity included in the model, as well as the source of multiple infections. This is outlined by both reviewers and these concerns must be addressed by the authors before the manuscript can be considered for publication in PLoS Computational Biology.

Similar to both of the reviewers comments on dissecting why there is agreement or lack of agreement between the simulated data and the two MOI approaches, the authors should consider:

(1) since the model appears to assume infection duration is determined solely by antigen variation and immune escape by var genes, the model should consider the possibility that infections also resolve randomly before exhaustion of the var gene repertoire (for example due to other mechanisms of infection control) - and whether this impacts on the performance of the MOI estimation.

(2) It is a little unclear but it seems the model assumes that sexual reproduction can occur but the source of donor parasites is always from a single host? If the model includes the possibility that multiple hosts can infect a single mosquitos, how will this impact the diversity and performance of the methods?

The editor suggests that the reviewers comments can be most adequately addressed by stress testing the structural assumptions of the model used for simulation (as outlined by Reviewer 1 and above), and/or by adapting a completely unique model of malaria transmission from the field and performing the same analysis using a different method of simulating data, and then confirming that the structure of the model does not dramatically impact the conclusions of the performance of each of the methods tested.

Reviewer's Responses to Questions

**Comments to the Authors:**

Reviewer #1: Neutral vs. non-neutral genetic footprints of Plasmodium falciparum multiclonal infections

In the submitted manuscript the authors use simulations to assess the relative performance of THE REAL MCCOIL and varcoding in estimating the true multiplicity of infection (MOI) of a malaria infection. The varcoding approach was previously developed by the authors in a prior publication, and THE REAL MCCOIL is a widely used approach independently developed. The issue of accurately capturing is highly pertinent for malaria control and elimination strategies, and for characterization of the epidemiology and intensity of malaria transmission. As such the authors use simulations which capture three broad levels of transmission intensity, and compare the measurement error in capturing MOI. Overall, the authors find utility in both approaches, though varcoding provides higher accuracy especially in higher transmission settings. While the comparison is very timely, and we are in need of these tools there were a number of issues which I feel were left unresolved in the current manuscript.

(1) Treatment of relatedness and comparability of simulated data. As the authors state in their introduction genetically distinct malaria parasites often co-infect individuals. These co-infecting parasite may come from either a single mosquito bite, multiple mosquito bites or both. This means a single individual may be infected with closely related parasites (i.e. siblings), unrelated parasites or a spectrum of the two. In the initial THE REAL MCCOIL paper intrahost relatedness was identified as a major impact on measurement accuracy. The simulation model includes meiotic recombination, though there is no information on intrahost relatedness and how it impacts estimates of MOI from either method. How many recombinants from the initial oocyst tetrad survive to the bloodstream of the donor infection? How many oocysts per mosquito are simulated?

It appears there is a fundamental issue in the way the data is simulated. Linkage between neutral markers is considered when generating recombinants, though not (seemingly) for var genes where a random set of types is selected from the parents. This may have profound implications for measurement error, especially at high MOI where varcoding shows greater benefit.

(2) The depth of simulations. The paper simulates three quite simplistic populations of malaria, albeit using a detailed and sophisticated model. These capture the steady state dynamics of high, medium and low transmission at the end of the wet season for seasonal malaria. While these are important categories, the utility of each approach to different settings is not explored in depth. For instance, in reality it is rare that there is instantaneous capture of parasite populations at a single time point (i.e. the end of the wet season), but continuous capture throughout a malaria season. The MOI under the model likely increases over a season, as does the mixing among the population of parasites. Does heterogenous sampling impact measurement error (also not all malaria transmission is seasonal, does this alter error?). Notably, these approaches will be used to characterize regions where transmission has changed. This may be due to a successful intervention, a rebound post-intervention or some other scenario. How does each approach perform in the presence of recently changed transmission intensity?

Reviewer #2: The review is uploaded as an attachment.

**Have the authors made all data and (if applicable) computational code underlying the findings in their manuscript fully available?**

Reviewer #1: Yes

Reviewer #2: None

PLOS authors have the option to publish the peer review history of their article (what does this mean?). If published, this will include your full peer review and any attached files.

Reviewer #1: No

Reviewer #2: No
---

## [Decision Letter · Decision Letter 1]

14 Dec 2022

Dear Dr. Pascual,

Thank you for your revised manuscript, and we note that in this revision you have made considerable improvements to addressed the reviewers comments.

We are pleased to inform you that your manuscript 'Neutral vs. non-neutral genetic footprints of Plasmodium falciparum multiclonal infections' has been provisionally accepted for publication in PLOS Computational Biology. 

Before your manuscript can be formally accepted there are some further minor suggested edits provided by reviewer 2 (see below), which we will leave to your discretion, and you will need to complete some formatting changes, which you will receive in a follow up email. A member of our team will be in touch with a set of requests.

Best regards,

David S. Khoury

Academic Editor

PLOS Computational Biology

Virginia Pitzer

Section Editor

PLOS Computational Biology

Reviewer's Responses to Questions

**Comments to the Authors:**

Reviewer #2: The authors clarified the questions I had. I have only two additional questions that can be addressed by adding the information to the text.

1. Line 679-686. THE REAL McCOIL gave the 95% credibility interval, mean, and median for the estimate of COI. Were median or mean used in this study? If the posterior distribution was not symmetric and the mean was used as the point estimate of COI, overestimation or underestimation may happen.

2. It's interesting that a higher proportion of minor single calls can lead to overestimation of COI. Was the estimated probability of calling double allele loci single allele loci higher than the actual error rate?

**Have the authors made all data and (if applicable) computational code underlying the findings in their manuscript fully available?**

Reviewer #2: None

PLOS authors have the option to publish the peer review history of their article (what does this mean?). If published, this will include your full peer review and any attached files.

Reviewer #2: No

---

## [Editor Report · Acceptance letter]

27 Dec 2022

PCOMPBIOL-D-22-01024R1 

Neutral vs. non-neutral genetic footprints of *Plasmodium falciparum* multiclonal infections

Dear Dr Pascual,

I am pleased to inform you that your manuscript has been formally accepted for publication in PLOS Computational Biology. Your manuscript is now with our production department and you will be notified of the publication date in due course.

With kind regards,

Zsofia Freund
